# Multi-label Co-regularization for Semi-supervised Facial Action Unit Recognition

**Xuesong Niu**[1,3]**, Hu Han**[1,2]**, Shiguang Shan**[1,2,3,4]**, Xilin Chen**[1,3]

[1] Key Laboratory of Intelligent Information Processing of Chinese Academy of Sciences (CAS),
Institute of Computing Technology, CAS, Beijing 100190, China
[2] Peng Cheng Laboratory, Shenzhen, China
[3] University of Chinese Academy of Sciences, Beijing 100049, China
[4] CAS Center for Excellence in Brain Science and Intelligence Technology, Shanghai, China
xuesong.niu@vipl.ict.ac.cn, {hanhu, sgshan, xlchen}@ict.ac.cn

## Abstract

Facial action units (AUs) recognition is essential for emotion analysis and has been widely applied in mental state analysis. Existing work on AU recognition usually requires big face dataset with accurate AU labels. However, manual AU annotation requires expertise and can be time-consuming. In this work, we propose a semi-supervised approach for AU recognition utilizing a large number of web face images without AU labels and a small face dataset with AU labels inspired by the co-training methods. Unlike traditional co-training methods that require provided multi-view features and model re-training, we propose a novel co-training method, namely multi-label co-regularization, for semi-supervised facial AU recognition. Two deep neural networks are used to generate multi-view features for both labeled and unlabeled face images, and a multi-view loss is designed to enforce the generated features from the two views to be conditionally independent representations. In order to obtain consistent predictions from the two views, we further design a multi-label co-regularization loss aiming to minimize the distance between the predicted AU probability distributions of the two views. In addition, prior knowledge of the relationship between individual AUs is embedded through a graph convolutional network (GCN) for exploiting useful information from the big unlabeled dataset. Experiments on several benchmarks show that the proposed approach can effectively leverage large datasets of unlabeled face images to improve the AU recognition robustness and outperform the state-of-the-art semi-supervised AU recognition methods. Code is available[1].

## 1 Introduction

Facial action units coded by the Facial Action Coding System (FACS) [5] refer to a set of facial muscle movements defined by their appearance on the face. These facial movements can be used for coding nearly any anatomically possible facial expression and have wide potential applications in mental state analysis, i.e., deception detection [10], diagnosing mental health [28], and improving e-learning experiences [22].

Most of the existing AU recognition methods are in a supervised fashion [3, 16, 21, 29, 40], for which a large number of facial images with AU labels are required. However, since AUs are subtle, local and have significant subject-dependent variations, qualified FACS experts are required to annotate facial AUs. In addition, labeling AUs is time-consuming and labor-intensive, making it impractical to

manually annotate a large set of face images. While at the same time, there are massive facial images, i.e., available on the Internet, video surveillance, social media, etc. There are still very limited studies about how to make such massive unlabeled face images to assist in AU recognition from a relatively small label face dataset.

As illustrated in Fig. 1(a), there could be different perspectives of the face that can be used for the classification of AUs. The diversity of different models trained for different views exists in both labeled and unlabeled face images and could further be used to enhance the generalization ability of each model. This idea is inspired by the traditional co-training methods [2], which has been proven to be effective for multi-view semi-supervised learning. However, traditional co-training methods usually require multiple views from different sources [2] or representations [20], which can be difficult to obtain in practice. In addition, traditional co-training methods usually need pseudo annotations for the unlabeled data and retraining the classifiers, which is not suitable for end-to-end training. Besides, traditional co-training methods seldom consider multi-label classification, which has its particular characteristics such as the correlations between different classifiers. Because of these reasons, co-training has seldom been studied for semi-supervised AU recognition. In recent years, deep neural networks (DNNs) have been proved to be effective for representation learning in various computer vision tasks, including AU recognition [3, 16, 21, 29]. The strong representation learning ability of DNNs makes it possible to generate multi-view representations that can be used for co-training based semi-supervised learning.

In addition, there exist strong correlation among different AUs. For example, as shown in Fig. 1(b), AU6 (cheek raiser) and AU12 (lip corner puller) are usually activated simultaneously in a common facial expression called Duchenne smile. There exist several methods that utilizing this prior knowledge to improve the AU recognition accuracy [1, 3, 6]. However, these methods are all in a supervised fashion, and their generalization abilities are limited by the sizes of existing labeled AU databases. At the same time, such correlations of AUs exist in face images not matter labeled or unlabeled, and could bring more robustness to the AU classifiers in a semi-supervised setting because more face images are considered.

In this paper, we propose a semi-supervised co-training approach named as multi-label co-regularization for AU recognition, aiming at improving AU recognition with massive unlabeled face images and domain knowledge of AUs. Unlike the traditional co-training method that requires provided multi-view features and model re-training, we propose a co-regularization method to perform semi-supervised co-training. For each facial image with or without AU annotation, we firstly generate features of different views via two DNNs. A multi-view loss $L_{mv}$ is used to enforce the two feature generators to get conditional independent facial representations, which are used as the multi-view features of the input image. A multi-label co-regularization loss $L_{cr}$ is also utilized to constrain the prediction consistency of two views for both labeled and unlabeled images. In addition, in order to leverage the AU relationships in both labeled and unlabeled images for our co-regularization framework, we use graph convolutional network (GCN) to embed the domain knowledge.

The contributions of this paper are as follows: i) we propose a novel multi-label co-regularization method for semi-supervised AU recognition leveraging massive unlabeled face images and a relatively small set of labeled face images; ii) we utilize the domain knowledge of AU relationships embedded via graph convolutional network to further mine the useful information of the unlabeled images; and iii) we achieve superior performance than the results without using massive unlabeled face images and the state-of-the-art semi-supervised AU recognition approaches.

## 2 Related Work

In this section, we review existing methods that are related to our work, including semi-supervised and weakly supervised AU recognition, AU recognition with relationship modeling, and co-training.

**Semi-supervised and Weakly supervised AU Recognition.** Previous works for semi-supervised AU recognition and weakly supervised AU recognition mainly focused on utilizing face images with incomplete labels, noisy labels or related emotion labels to improve the AU recognition accuracy. Wu et al. [33] proposed to use Restricted Boltzmann Machine to model the AU distribution, which is further used to train the AU classifiers with partially labeled data. Peng et al. [26] proposed an adversary network to improve the AU recognition accuracy with emotion labeled images. In [39], Zhao et al. proposed a weakly supervised clustering method for pruning noise labels and trained

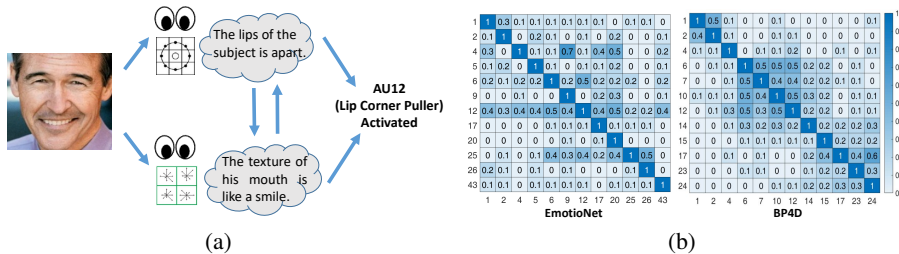

(a)                                                                (b)

Figure 1: (a) An illustration of the idea of co-training. For an input image, representations generated by different models can highlight different cues for AU recognition. Sharing such kind of multi-view representations for unlabeled images can improve the generalization ability of each model. (b) The correlation maps of different AUs calculated based on Equ. 8 in Section 3.4 for EmotioNet and BP4D databases suggest that there exist strong correlations between different AUs.

the AU classifiers with re-annotated data. Although these methods do not need massive complete AU labeled images, they still need other annotations such as noise AU labels or emotion labels as well as labeled face images. Recently, several methods tried to utilize the face images with only emotion labels to recognize AUs. Peng et al. [25] utilized the prior knowledge of AUs and emotions to generate pseudo AU labels for training from facial images with only emotion labels. Zhang et al. [37] proposed a knowledge-driven strategy for jointly training multiple AU classifiers without any AU annotation by leveraging prior probabilities on AUs. Although these methods do not need any AU labels, they still need the related emotion labels. Besides, most of these methods are evaluated on lab-collected data, and their generation abilities to the web facial images are limited.

**AU Recognition based on Relationship Modeling.** There exist strong probabilistic dependencies between different AUs that can be treated as domain knowledge and further used for AU recognition. Almaev et al. [1] exploited to learn the classifier for one single AU and transfer the classifier to other AUs using the latent relations between different AUs. Eleftheriadis et al. [6] proposed a latent space embedding method for AU classification by considering the AU label dependencies. In [3], Corneanu et al. proposed a structure inference network to model AU relationships based on a fully-connected graph. However, all these methods need fully annotated face images, and the generalization abilities of these models are limited because of the limited sizes of existing AU databases.

**Co-training for Semi-supervised Learning.** Semi-supervised learning is a widely studied problem, and many milestone works have been proposed, i.e., Mean-teacher method [31], Transductive SVM [14], etc. Among these works, co-training [2] is designed for multi-view semi-supervised learning and has been proved to have good theoretical results. Traditional co-training methods are mainly based on provided multi-view data, which is usually not available in practice. Meanwhile, they usually need to get the pseudo labels for retraining, making it impractical for end-to-end training. Recently, Qiao et al. [27] utilized the adversarial examples to learning the multi-view features for multi-class image classification. However, for the problem of facial AU recognition, it is hard to get the adversarial examples for multiple classifiers. In [34], Xing et al. proposed a multi-label co-training (MLCT) method considering the co-occurrence of pairwise labels. However, they still required provided multi-view features for training, which may be hard to obtain.

## 3 Proposed Method

In this section, we first introduce the traditional co-training. Then, we detail the proposed multi-label co-regularization approach for AU recognition with AU relationship learning.

### 3.1 Traditional Co-training

The traditional co-training approach [2] is an award-winning method for semi-supervised learning. It assumes that each sample in the training set has two different views $v_1$ and $v_2$, and each view can provide sufficient information to learn an effective model. The two different views in the co-training assumption are from different sources or data representations. Two models, i.e., $M_1$ and $M_2$, are trained based on $v_1$ and $v_2$, respectively. Then the predictions of each model for the unlabeled data are

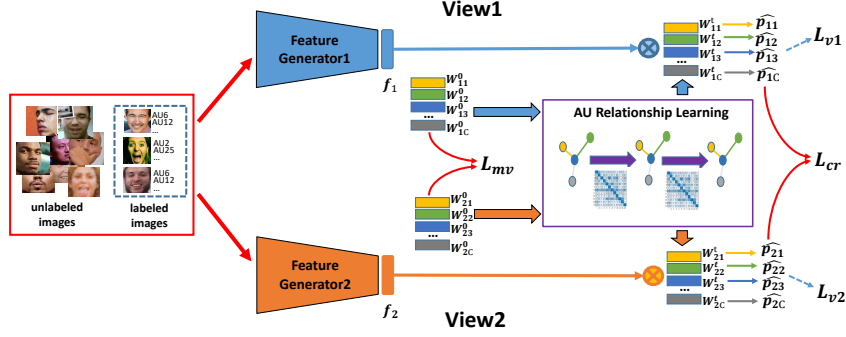

Figure 2: An overview of the proposed multi-label co-regularization method for semi-supervised AU recognition. The losses defined for labeled facial images, i.e., $L_{v1}$ and $L_{v2}$, are illustrated with blue dash lines. The losses defined for both the labeled and unlabeled images, i.e., $L_{mv}$ and $L_{cr}$, are illustrated using red solid lines.

used to augment the training set of the other model. This procedure is conducted for several iterations until $M_1$ and $M_2$ become stable. This simple but effective approach can significantly improve the models' performance when it is used to exploit useful information from massive unlabeled data, and has been proven to have PAC-style guarantees on semi-supervised learning under the assumption that two views are conditionally independent [2]. There are two characteristics that guarantee the success of co-training: i) the two-view features are conditionally independent; ii) the models trained on different views tend to have similar predictions because of the re-training mechanism. Our multi-label co-regularization method is designed based on these two characteristics.

## 3.2 Multi-view Feature Generation

Deep neural networks have been proved to be effective in feature generation [11, 13, 30]. We utilize two deep neural networks to generate the two-view features [4]. Here, we choose two ResNet-34 networks [13] as the feature generators. Given a facial image dataset $D = L \bigcup U$, where $L$ denotes the facial images with AU labels and $U$ denotes the unlabeled face images. For each image in $D$, the two-view features $f_1$ and $f_2$ can be generated using the two different generators. Then $C$ classifiers can be learned to predict the probabilities of $C$ AUs using the feature of each view. Let $\hat{p_{ij}}$ denotes the probability predicted for the $j$-th AU using the $i$-th view, and the final probabilities can be formulated as

$$\hat{p_{ij}} = \sigma(w_{ij}^T f_i + b_{ij}); \tag{1}$$

where $\sigma$ denotes the sigmoid function, and $w_{ij}$ and $b_{ij}$ are the respective classifier parameters. We first calculate the losses for all the labeled images in $L$. A binary cross-entropy loss is utilized for both view. In order to better handle the data imbalance [32, 23] of AUs, a selective learning strategy [12] is adopted. The loss function for AU recognition for the $i$-th view $L_{vi}$ is formulated as

$$L_{vi} = -\frac{1}{C} \sum_{j=1}^{C} a_c[p_j \log \hat{p_{ij}} + (1 - p_j) \log(1 - \hat{p_{ij}})] \tag{2}$$

where $p_j$ is the ground-truth probability of the occurrence for the $j$-th AU, with 1 denoting occurrence of an AU and 0 denoting no occurrence. $a_c$ is a balancing parameter calculated in each batch using the selective learning strategy [12].

One key characteristic of co-training is the input multi-view features are supposed to be conditional independent. Although different networks may achieve similar performance in a complementary way when they are initialized differently, they can gradually resemble each other when they are supervised by the same target. In order to encourage the two feature generators to get conditional independent features instead of collapsing into each other, we proposed a multi-view loss by orthogonalizing the weights of the AU classifiers of different views. The multi-view loss $L_{mv}$ is defined as

$$L_{mv} = \frac{1}{C} \sum_{j=1}^{C} \frac{W_{1j}^T W_{2j}}{\|W_{1j}\| \|W_{2j}\|} \tag{3}$$

where $W_{ij} = [w_{ij} \ b_{ij}]$ denote the parameters of the $j$-th AU's classifier of the $i$-th view. With this multi-view loss, the features generated for different views are expected to be different while complementary with each other.

### 3.3 Multi-label Co-regularization

Besides of the conditional independent assumption, another key characteristic of co-training is to force the classifiers of different views to get consistent predictions. Instead of using the labeling and re-training mechanism in the traditional co-training methods, we propose a co-regularization loss to encourage the classifiers from different views to generate similar predictions. For the face images in $D$, we first get the predicted probabilities $\hat{p_{ij}}$ for the $j$-th AU from the classifier of the $i$-th view. Then, we try to minimize the distance between the two predicted probability distributions w.r.t. all AU classes from the two views. The Jensen-Shannon divergence [7] is utilized to measure the distance of two distributions, and the co-regularization loss is defined as

$$L_{cr} = \frac{1}{C} \sum_{j=1}^{C} (H(\frac{\hat{p_{1j}} + \hat{p_{2j}}}{2}) - \frac{H(\hat{p_{1j}}) + H(\hat{p_{2j}})}{2}) \tag{4}$$

where $H(p) = -(p \log p + (1-p) \log(1-p))$ is the entropy w.r.t $p$. The final loss function of our multi-label co-regularization can be formulated as

$$L = \frac{1}{2} \sum_{i=1}^{2} L_{vi} + \lambda_{mv} L_{mv} + \lambda_{cr} L_{cr} \tag{5}$$

where $\lambda_{mv}$ and $\lambda_{cr}$ are hyper-parameters that balance the influences of different losses.

### 3.4 AU Relationship Learning

Based on the nature of facial anatomy, there exist strong relationships among different AUs. In order to make full use of such correlation as prior knowledge existing in massive unlabeled images, we further embed such prior knowledge into our multi-label co-regularization AU recognition model via graph convolutional network (GCN) [15]. GCN has been an effective model for message passing between different nodes. In this paper, we use a two-layer GCN. The parameters $W_{ij}$ of AU classifiers for both views are used as the nodes of GCN. We formulate the input and output of GCN in the format of $W_i = [W_{i1}; W_{i2}; \cdots; W_{iC}]$, where the $j$-th column of $W_i$ is the parameters of the classifier for $j$-th AU. The message passing mechanism of GCN can be formulated as

$$W_i^t = \hat{A} \ ReLU(\hat{A} W_i^0 H^{(0)}) H^{(1)}; \tag{6}$$

where $W_i^0$ and $W_i^t$ denote the input and output of GCN, respectively; $\hat{A}$ is the adjacency matrix of the nodes; $H^{(0)}$ and $H^{(1)}$ are the parameters of GCN. $ReLU$ is the activation function for GCN, and we use Leaky ReLU [35]. We first pre-train the feature generators and classifiers with $L_{vi}$, $L_{mv}$ and $L_{cr}$ with all the labeled and unlabeled images. Then we use the weights of the pre-trained classifiers as the initial input of GCN (i.e., $W_i^0$) and fine-tune the network and GCN on both labeled and unlabeled images.

The key part of GCN is to define the adjacency matrix $\hat{A}$. Here, we choose to use the dependency matrix calculated from the labeled data as the adjacency matrix. Since there are only a small number of positive samples for some AUs, we consider the dependency for both positive and negative samples to reduce the influence of imbalance. We first compute the average dependency matrix for $C$ AUs as

$$P_{dep} = \frac{1}{2} ([P(L_i = 1 | L_j = 1)]_{C \times C} + [P(L_i = -1 | L_j = -1)]_{C \times C}) \tag{7}$$

Since a dependency probability $P(L_i = 1 | L_j = 1) = 0.5$ means that when $j$-th AU is activated, the probability of occurrence is equal to the no occurrence for $i$-th AU. This indicates that the activation of $j$-th AU could not provide useful information for the $i$-th AU; thus there should be no link between the two nodes. At the same time, all the elements of the adjacency matrix should be positive, and the diagonal should be 1; therefore, we further modify the $P_{dep}$ and calculate the adjacency matrix as

$$\hat{A} = ABS((P_{dep} - 0.5) \times 2) \tag{8}$$

where ABS(M) returns a matrix whose elements are the absolute value of the elements of M.

After training the two view AU recognition networks with our multi-label co-regularization method and AU relationship learning, the two networks tend to get more representative features for AU classification, and the prior knowledge of AU relationship is also embedded in the AU classifiers. With these robust AU features and classifiers learned from labeled and unlabeled face images, we could significantly improve the AU recognition accuracy.

## 4 Experimental Results

### 4.1 Experimental Settings

#### 4.1.1 Databases and Protocols

We provide evaluations on two widely used AU databases, i.e., EmotioNet [8] and BP4D [36].

**EmotioNet** is an in-the-wild database for facial AU recognition containing about 1M images downloaded from the Internet. 20,722 face images were provided with manual annotations for 12 AUs by experts. The face images without manual annotations are used as the unlabeled training dataset. We randomly choose 15,000 manually annotated images as the labeled training set, and the other images are used for testing. In order to reduce the bias of a single random dataset split, we perform the testing three times and report the average performance of the three tests.

**BP4D** is a spontaneous facial AU database containing 328 videos from 41 subjects. Each subject is involved in 8 sessions, and the spontaneous facial expressions are recorded. 12 AUs are annotated for all the video frames, and there are about 140,000 images with AU labels. For all the experiments on BP4D, we conduct a subject-exclusive 3-fold cross-validation test following [16, 29]. The unlabeled training images used for experiments on BP4D are taken from the EmotioNet database.

For all the images used in the experiments, we utilize an open source SeetaFace[2] face detector to detect the face and five facial landmarks. All the facial images are then aligned and cropped to $240 \times 240$ based on the five facial landmarks. The aligned face images are randomly cropped to $224 \times 224$ for network input during training. Images center-cropped from the aligned face images are utilized for testing. Random horizontal flipping is used for data augmentation.

#### 4.1.2 Training Details

We incrementally train our multi-label co-regularization method. Two ResNet-34 models are chosen as the feature generators, and the fully connected layers of the two feature generators are regarded as the multi-label classifiers. For the experiments on EmotioNet and BP4D, we first pre-train the two feature generators by setting $\lambda_{mv} = 400$ and $\lambda_{cr} = 100$. The Adam optimizer with an initial learning rate of 0.001 is applied to optimize the feature generators and AU classifiers. Then, we take GCN into account and jointly train the feature generators and GCN. The initial learning rate is set to 0.001 for GCN and 0.0001 for the two feature generators. We remove $\lambda_{mv}$ when training GCN since the feature generators could already provide multi-view features after pre-training. The parameters of GCN are shared between two views. We set the maximum iteration to 60 epochs for pre-training the feature generators and 20 epochs for fine-tuning the GCN and feature generators. The batch size for all the experiments is set to 100. The numbers of unlabeled face images for experiments are 50,000 and 100,000 for EmotioNet and BP4D datasets, respectively. During each epoch, we first combine the labeled and unlabeled images and sample the face images randomly to train the model. Then we optimize the model with only the labeled face images. All the experiments are conducted with PyTorch [24] on a GeForce GTX 1080 Ti GPU.

#### 4.1.3 Evaluation Metrics

Following the previous methods for AU recognition [3, 21, 40], we use F1 score for all the experiments. We also report the average F1 score over all AUs (denoted as Avg.). For fair comparisons with the baseline methods, we choose the **average performance** of the two views as the final performance of our method.

Table 1: F1 score (in %) for recognition of 12 AUs by the proposed method and the state-of-the-art semi-supervised methods on the EmotioNet database.

| AU<br>Method | 1 | 2 | 4 | 5 | 6 | 9 | 12 | 17 | 20 | 25 | 26 | 43 | Avg. |
|---|---|---|---|---|---|---|---|---|---|---|---|---|---|
| Baseline | 55.6 | 41.1 | 70.1 | 46.6 | 80.2 | 59.2 | 90.7 | 45.5 | 45.1 | 94.2 | 58.7 | 60.5 | 62.3 |
| MLCT [40] | 57.8 | 44.8 | 73.7 | 50.1 | 82.8 | 58.1 | 91.8 | 44.8 | 37.1 | 95.1 | 61.6 | 63.4 | 63.4 |
| Mean-teacher [38] | 55.5 | 46.3 | 71.1 | 48.6 | 81.6 | 61.7 | 91.0 | 46.7 | 43.5 | 94.7 | 60.2 | 63.9 | 63.7 |
| Co-training [9] | 58.3 | 48.4 | 70.0 | 50.4 | 83.1 | 64.4 | 91.7 | 49.9 | 47.1 | 95.0 | 60.0 | 66.9 | 65.5 |
| Proposed | **61.4** | **49.3** | **75.9** | **54.1** | **83.5** | **68.3** | **92.0** | **50.8** | **53.5** | **95.2** | **65.1** | **68.1** | **68.1** |

Table 2: F1 score (in %) for recognition of 12 AUs by the proposed method and the state-of-the-art methods on the BP4D database. The unlabeled images in this experiment are from the EmotioNet dataset.

| AU<br>Method | 1 | 2 | 4 | 6 | 7 | 10 | 12 | 14 | 15 | 17 | 23 | 24 | Avg. |
|---|---|---|---|---|---|---|---|---|---|---|---|---|---|
| Baseline | 41.7 | 31.0 | 47.9 | 75.2 | 76.9 | 80.0 | 85.5 | 60.3 | 35.9 | 58.5 | 37.6 | 47.8 | 56.5 |
| Mean-teacher [38] | 40.6 | **40.7** | 47.7 | 76.2 | **77.9** | 80.7 | 85.8 | 61.1 | 33.6 | 62.3 | 41.9 | 44.6 | 57.8 |
| Co-training [9] | **44.1** | 39.9 | 46.9 | 75.2 | 77.3 | 81.6 | **86.2** | **61.5** | 35.4 | 62.7 | 39.6 | 45.3 | 58.0 |
| Proposed | 42.4 | 36.9 | **48.1** | **77.5** | 77.6 | **83.6** | 85.8 | 61.0 | **43.7** | 63.2 | 42.1 | **55.6** | **59.8** |
| ROI [16] | 36.2 | 31.6 | 43.4 | 77.1 | 73.7 | 85.0 | 87.0 | 62.6 | 45.7 | 58.0 | 38.3 | 37.4 | 56.4 |
| JAA-Net [29] | 47.2 | 44.0 | 54.9 | 77.5 | 74.6 | 84.0 | 86.9 | 61.9 | 43.6 | 60.3 | 42.7 | 41.9 | 60.0 |

## 4.2 Results on EmotioNet

Three state-of-the-art semi-supervised methods (traditional co-training [2], mean-teacher [31] and multi-label co-training (MLCT) [34]) are used for comparison on EmotioNet. For traditional co-training [2] and mean-teacher [31], we choose the same backbone network (ResNet-34) for our approach for fair comparison. For MLCT [34] which needs provided multi-view features, we utilize the 512-D features extracted by two ResNet-34 networks trained on EmotioNet. All the results are shown in Table 1. The performance trained by ResNet-34 using only the annotated face images (denoted as Baseline) is also provided in Table 1.

From the results, we can see that when comparing with the ResNet-34 trained using only labeled data, all the semi-supervised methods can improve the performance by exploiting useful information from unlabeled face images. When comparing with MLCT [40], which requires provided multi-view features, our method outperforms MLCT [40] by a large margin on all the AUs and the average F1 score. This indicates that optimizing feature learning along with the classifiers is helpful for improving the AU recognition accuracy for semi-supervised learning. When comparing with mean-teacher [38], which utilizes a single network to learn the features with both labeled and unlabeled images, our method outperforms it by a large margin, indicating that more AU discriminative information could be obtained from the unlabeled face images through multi-view feature generation. In addition, our method also outperforms the traditional co-training method [9], suggesting that the proposed multi-label co-regularization could provide a better way in exploiting the two-view information through end-to-end training and thus improve the generalization ability.

## 4.3 Results on BP4D

We compare the proposed approach with two state-of-the-art semi-supervised learning methods (traditional co-training [2] and mean-teacher [31]) on the BP4D database. The results of the model trained only with labeled face images (denoted as Baseline) are also provided. Besides, BP4D database is the largest face images database with AU labels, and many meticulously-designed supervised AU recognition systems have been evaluated on this database. We choose two state-of-the-art supervised AU recognition systems (ROI [16] and JAA-Net [29]) for comparison and their performance are taken from [29]. All the results are reported in Table 2.

From the results, we can see that the proposed semi-supervised method could significantly improve the AU recognition accuracy and outperforms the state-of-the-art semi-supervised methods, i.e.,

Table 3: F1 score (in %) for recognition of 12 AUs in the ablation study on the EmotioNet database.

| Method \ AU | 1 | 2 | 4 | 5 | 6 | 9 | 12 | 17 | 20 | 25 | 26 | 43 | Avg. |
|---|---|---|---|---|---|---|---|---|---|---|---|---|---|
| Baseline | 55.6 | 41.1 | 70.1 | 46.6 | 80.2 | 59.2 | 90.7 | 45.5 | 45.1 | 94.2 | 58.7 | 60.5 | 62.3 |
| Baseline+$L_{cr}$ | 61.7 | 47.1 | 74.7 | 53.3 | 82.9 | 66.1 | 91.8 | 50.5 | 51.9 | 95.2 | 63.4 | 66.4 | 67.1 |
| Baseline+$L_{cr}$+$L_{mv}$ | 61.7 | 47.4 | 74.7 | 53.9 | 82.9 | 67.5 | 91.7 | 51.7 | 53.0 | 95.0 | 63.6 | 66.9 | 67.5 |
| Baseline+$L_{cr}$+$L_{mv}$+GCN | 61.4 | 49.3 | 75.9 | 54.1 | 83.5 | 68.3 | 92.0 | 50.8 | 53.5 | 95.2 | 65.1 | 68.1 | 68.1 |

co-training [2] and mean-teacher [38]. When comparing with ROI [16] and JAA-Net [29], our method could achieve better or comparable average performance using a general RenNet-34 network without specific designs and unlabeled face images. By contrast, both JAA-Net [29] and ROI [16] have used additional features such as facial landmarks, which are helpful for the recognition of AUs defined in small regions, i.e., AU1 and AU2. In addition, the unlabeled facial images are from the Internet, which are very different from the labeled face images in BP4D in terms of pose, illumination and background variations, introducing additional challenges for semi-supervised AU recognition.

## 4.4 Ablation Study

We provide ablation study to investigate the effectiveness of the key components ($L_{mv}$, $L_{cr}$ and GCN) of the proposed multi-label co-regularization method. We add them step by step to the model trained only with labeled face images (denoted as Baseline). The results of the ablation study are provided in Table 3.

**Effectiveness of $L_{mv}$ and $L_{cr}$.** From the results in Table 3, we can see that the improvement is mainly gained from the $L_{cr}$, with an average F1 score increased from 62.3% to 67.1%, which indicates the effectiveness of our multi-label co-regularization. Even when $L_{mv}$ is not used, two networks may generate different features if they are initialized differently. When we use $L_{mv}$, the average F1 can be further improved to 67.5%.

In order to validate that $L_{mv}$ is useful for the two networks to learn different but complementary features, we further ensemble the two networks with an average of their predicted probabilities. We notice that ensemble achieves an average F1 score of 67.2% without $L_{mv}$, which is close to the result without ensemble. When $L_{mv}$ is used, the ensemble can obtain an average F1 of 68.1%, a higher result than that without ensemble (67.5% average F1 score). In addition, we also calculate the proportion of samples with inconsistent predictions from the two views with and without $L_{mv}$, and the results are shown in Fig. 3(a). These results indicate that more diverse representations could be learned with the help of $L_{mv}$, and thus improve the AU recognition performance of our method.

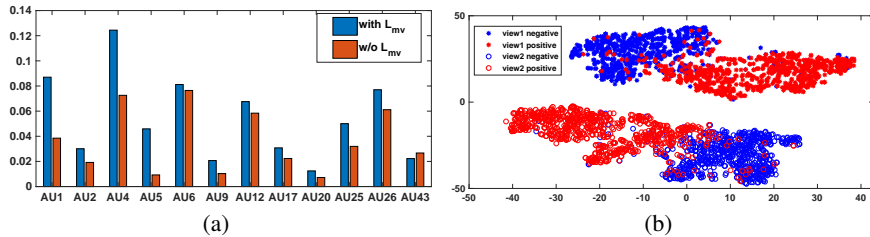

(a)  (b)

Figure 3: (a) Proportion of samples with inconsistent prediction results from the two views. (b) t-SNE visualization of the features generated from two views to recognize AU25.

Besides the quantitative results, we also use t-SNE [19] to visualize the features generated from the two views to recognize AU25 in Fig. 3(b). From the results, we can see that both views can achieve good classification accuracies, and the features generated from different views are very different. This indicates that the two networks do learn different cues for AU recognition. From the results, we can see that orthogonalizing weights can make the predictions of the two views more diverse, and thus benefit the semi-supervised co-training.

**Effectiveness of GCN.** We finally include GCN to our multi-label co-regularization method. The average F1 score is further improved to 68.1% from 67.5%. For the AUs that are coexistent or exclusive with other AUs, such as AU4 and AU9, the performance improvement is more evident, i.e.,

from 47.4% and 67.5% to 49.3% and 68.3%, respectively. This indicates that GCN can effectively leverage the correlations between individual AUs to improve the AU recognition accuracies.

Considering that the AU co-occurrences may be different for different databases, we further conduct cross-database evaluations to see whether exploiting AU relationships could provide better generalization ability for AU recognition. We first train our model on EmotioNet with and without GCN, and then test the models on BP4D and UNBC-McMaster [18]. UNBC-McMaster is a database for pain detection containing 48,398 FACS coded frames. AUs coded on both EmotioNet and the testing database are considered, and the results are given in Table 4. From the results we can see that our model using GCN for AU relationship modeling could provide better generalization ability.

Table 4: F1 score (in %) for cross-database testing on BP4D and UNBC-McMaster using models trained on EmotioNet with and without using GCN for AU relationship modeling.

| AU | BP4D | | | | | | | UNBC-McMaster | | | | | | | | |
|---|---|---|---|---|---|---|---|---|---|---|---|---|---|---|---|---|
| | 1 | 2 | 4 | 6 | 12 | 17 | Avg. | 4 | 6 | 9 | 12 | 20 | 25 | 26 | 43 | Avg. |
| w/o GCN | 24.3 | 18.5 | 23.1 | 56.8 | **65.5** | 27.8 | 36.0 | 82.4 | 53.6 | 42.1 | 56.6 | 32.8 | 87 | 19.2 | 71.6 | 55.6 |
| with GCN | **36.2** | **27.8** | **26.4** | **60.1** | 65.4 | **32.4** | **41.4** | **91.2** | **62.5** | **70.4** | **67.0** | **53.2** | **89.6** | **41.9** | **74.4** | **68.8** |

## 4.5 Generalization Ability

If we treat facial AU recognition as a special kind of facial attributes, it would be interesting to see how the proposed semi-supervised learning approach generalizes to other face attribute estimation tasks. Therefore, we perform facial attributes estimation on the CelebA database [17]. CelebA is a large-scale facial attribution database containing 202,599 face images with 40 binary attribute annotations. We randomly choose 30,000 images as the labeled training set and 10,000 images as the test set. We choose 100,000 face images from the other face images as unlabeled face images. Again, we also repeat the experiments three times and report the average results. The baseline network remains ResNet-34. The F1 scores for all the forty attributes as well as the average F1 score are given in Fig. 4. We can see that the proposed approach can obtain improved face attribute estimation performance for all the 40 attributes when using the unlabeled face dataset for semi-supervised learning. This experiment suggests that our method has good generalization ability into similar tasks.

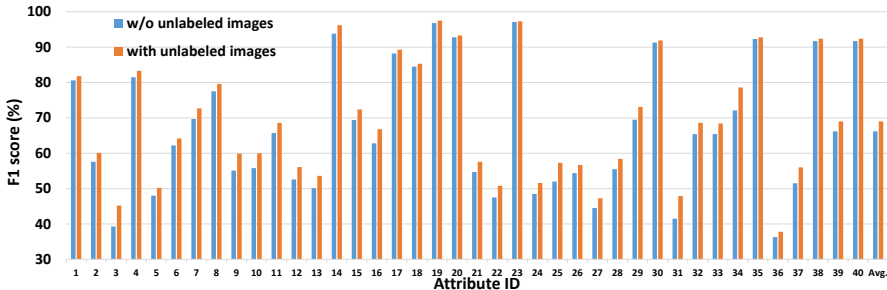

Figure 4: F1 score (in %) for the 40 attributes estimation by the proposed multi-label co-regularization approach with and without using massive unlabeled face images.

## 5 Conclusion

In this paper, we proposed a novel multi-label co-regularization method for semi-supervised facial AU recognition via co-training. Unlike the traditional co-training framework that needs provided multi-view features and re-training mechanism for the unlabeled data, the proposed approach enables jointly optimize multi-view feature generation and AU classification via multi-view loss and multi-label co-regularization loss. AU relationships are also embedded using GCN for exploiting more AU informative information from the unlabeled face images. The proposed approach achieves superior AU recognition performance than the state-of-the-art semi-supervised learning methods. Experiments for facial attribute estimation reveal good generalization ability of the proposed approach into the other tasks. In our further work, we would like to explore the use of weakly annotated face images from the Internet for AU recognition.

## Acknowledgement

This research was supported in part by the National Key R&D Program of China (grant 2017Y-FA0700800), Natural Science Foundation of China (grants 61672496 and 61702481), External Cooperation Program of Chinese Academy of Sciences (CAS) (grant GJHZ1843), and Youth Innovation Promotion Association CAS (2018135).

## Footnotes

[1] https://github.com/nxsEdson/MLCR

[2] `https://github.com/seetaface/SeetaFaceEngine`

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
