[Reviews · NeurIPS 2019]

Reviewer 1



The writing of this paper is clear. The motivations of each designed loss function and the term are explained well. There are some questions I would like the authors to answer: 1. As far as I know, this submission is not the first work to introduce GCN for taking advantages of prior knowledge in Action Units. Please refer to this work "Semantic Relationships Guided Representation Learning for Facial Action Unit Recognition". I would appreciate it if there is a comparison with this work in the rebuttal, which can highlight the contributions. 2. This work achieves a 59.8 Avg on BP4D, but "Semantic Relationships Guided Representation Learning for Facial Action Unit Recognition" achieves 62.9. The performance gap seems a little large, doesn't it? 3. In the submission, it said that "dependency matrix" is utilized and therefore it gives me a feeling that only dependency relation is considered here. The mutually exclusive relations between Action Units are not considered here? But from Equation 8, it seems mutual exclusive relations are also considered. (Please correct me if I am wrong at this place.) 4. Lmv term, which orthogonalizes the weights between different classifiers, is not proposed in this work for the first time. As far as I know, the work "Taking A Closer Look at Domain Shift: Category-level Adversaries for Semantics Consistent Domain Adaptation" already proposes to use this. Please check equation 4 of that work.

Reviewer 2



The paper explores a method for exploiting multi-view training with label co-regularization for facial action unit recognition. Strengths: 1. A method for exploiting unlabeled data for the task of action unit recognition which is consistently data poor, so such a method could contribute a lot to the field. 2. A method for better multi-view training through orthogonalization and co-regularization of features 3. Promising results on standard datasets Weaknesses: 1. One major risk of methods that exploit relationships between action units is that the relationships can be very different accross datasets (e.g. AU6 can occur both in an expression of pain and in happiness, and this co-occurence will be very different in a positive salience dataset such as SEMAINE compared to something like UNBC pain dataset). This difference in correlation can already be seen in Figure 1 with quite different co-occurences of AU1 and AU12. A good way to test the generalization of such work is by performing cross-dataset experiments, which this paper is lacking. 2. The language in the paper is sometimes conversational and not scientific (use of terms like massive), and there are several opinions and claims that are not substantiated (e.g. "... facial landmarks, which are helpful for the recognition of AUs defined in small regions"), the paper could benefit from copy-editing 3. Why are two instances of the same network (resnet) are used as different views? Would using a different architecture instead be considered a more differing view? Would be great to see a justification for using two resnet networks. 4. Why is the approach limited to two views, it feels like the system should be able to generalize to more views without too much difficulty? Minor comments: - What is PCA style guarantee? - What is v in equation 2? - why are dfferent numbers of unlabeled images using in training BP4D and EmotioNet models? Trivia: massive face images -> large datasets donates -> denotes (x2) adjacent -> adjacency

Reviewer 3



Good Points: 1) Fairly organized. 2) Solid Experiment. 3) Ablations studies. Question to the authors: 1) Authors argued that two different networks learn different cues for AU recognition in Fig 1, however did not provide any solid evidence of their intuition. i.e., feature visualization or examples which can be correctly classified by one but not by the other because of different cues learned and utilized. 2) Authors argued using orthogonal weights for the last layer makes the feature generator conditionally independent. Why? 3) What is the baseline for table 1 and 2. If it is a single network, I am not convinced it is a fair comparison. The baseline should have comparable number of parameters with the proposed method, may be two resnets trained separately. 4) More supervised methods could be added in the tables 1,2 along with JAA-Net, which will give readers a better idea on the effectiveness of semisupervised methods. 5) There are some typos, e.g., in line 261, donated --> denoted.

[Author Response · NeurIPS 2019]

We thank all the reviewers for their valuable suggestions. Our response to individual reviewers' concerns are as follows.

**======To Reviewer 1======**

(1) **The differences between our work and [1]** include: (i) *The scope of the two papers is different.* While [1] is a fully supervised action unite (AU) recognition method, we focus on *semi-supervised* using massive unlabeled data and a small set of labeled data, which is more challenging but meaningful since labeling AU is difficult/expensive. (ii) The usage of AU relationship is different. While [1] used GNN to integrate the semantic relationship between AUs to enhance feature representation, in our work, leveraging GCN to encode AU relationship prior is only one part of our work, which can benefit the key part of our work, mining useful information from massive unlabeled data to obtain more informative and generalizable representation than learning from only a small labeled dataset. (iii) The performance on BP4D and DISFA. The performance of our semi-supervised learning method is lower than [1] on BP4D. However, we further conduct experiments on DISFA as [1] (with 100K unlabeled images from EmotioNet), and our method achieves 56.8% Avg. F1 score, which is *higher* than that of [1] (55.9%). Although our semi-supervised learning method does not outperform the supervised learning method [1] on both datasets, we can still see the big potential of semi-supervised learning in AU recognition.

(2) **Consideration of mutually exclusive relations.** Our method also models the mutually exclusive relations of AUs. If two AUs are mutually exclusive, the avg. probability calculated by Eq. 7 will be small, and after normalization using Eq. 8, there will be a link added to the two AUs in the adjacent matrix.

(3) **Novelty of $L_{mv}$.** Our method learns two diverse classifiers in order to exploit diverse and informative features from unlabeled data for semi-supervised multi-label classification. Although the suggested CVPR19 paper also learns two diverse classifiers with paired labeled data for segmentation, the purpose is to predict how well each feature is semantically aligned between the source and target domains.

**======To Reviewer 2======**

(1) **Cross-database testing.** For cross-database testing, we train our model on EmotioNet with and without GCN and test the models on BP4D and UNBC (see Fig. 1(a)). Therefore, we agree that the AU co-occurrences may be different for different databases, but exploiting AU relationships provides better robustness of generated features for semi-supervised AU recognition under cross-database testing scenarios.

(2) **Language.** We will use copy-editing to improve our writing in the final version.

(3) **Justification for using two ResNet networks.** We conduct another experiment using ResNet-34 and Inception-v3 Network, instead of using two ResNets. The avg. F1-score on EmotioNet is 67.6%, which is similar to using two ResNet-34 (68.1% F1 score). The results indicate that using two ResNets can generate features of two views which are different enough from each other. The main reasons are: (i) the two ResNets are initialized differently (pretrained separately); (ii) we have utilized $L_{mv}$ to enforce them to generate different features.

(4) **Generalization to more views.** Currently, the $L_{cr}$ and $L_{mv}$ losses are designed for two views; one way to generalize to more views is to apply $L_{cr}$ and $L_{mv}$ to every two views. We will study this in future work.

(5) **Answers to the minor comments.** (a) PAC is a framework for mathematical analysis of machine learning, aiming at getting low generalization error with high probability. (b) "v" in Equ. 2 stands for "view". (c) We choose the number of unlabeled images according to the sizes of databases.

**======To Reviewer 3======**

(1) **Evidence of that two different networks can learn different cues for AU recognition.** We use t-SNE to visualize the features generated from the two views to recognize AU25 in Fig. 1(b). From the results, we can see that both views can achieve good classification accuracies, and the features generated from different views are very different, indicating that the two networks do learn different cues for AU recognition.

(2) **Explanation of the benefit of orthogonal weights.** Theoretically, the classifier with weights $w$ and input feature $f$ can be formulated as $\sigma(w^T f)$. After the model converges, the directions of vector $w$ and $f$ tend to be the same when the label is positive and tend to be opposite when the label is negative. So, $w$ can be regarded as a representation of all the learned features. Therefore, orthogonalizing the weights will make the classifier weights independent to each other, and thus lead to the generated features from different views to be conditional independent because the feature generators and classifiers are optimized together. The feature visualization in Fig. 1(b) can also verify this conclusion. In addition, we also calculate the proportion of samples with inconsistent predictions from the two views with and without $L_{mv}$, and the results are shown in Fig. 1(c). From the results, we can see that orthogonalizing weights can make the predictions of the two views more different, and thus further benefit the semi-supervised co-training.

(3) **Fair comparison with the baseline model.** For fair comparisons, we use the *average F1 score* of the two ResNets without ensembling them as the final performance for both baseline and the proposed method, which guarantees that the proposed method is compared with the baseline under the same scale of parameters.

| BP4D | | | | | | | |
|---|---|---|---|---|---|---|---|
| AU | 1 | 2 | 4 | 6 | 12 | 17 | Avg. |
| w/o GCN | 24.3 | 18.5 | 23.1 | 56.8 | 65.5 | 27.8 | 36.0 |
| with GCN | 36.2 | 27.8 | 26.4 | 60.1 | 65.4 | 32.4 | 41.4 |

| UNBC | | | | | | | | |
|---|---|---|---|---|---|---|---|---|
| AU | 4 | 6 | 9 | 12 | 20 | 25 | 26 | 43 | Avg. |
| w/o GCN | 82.4 | 53.6 | 42.1 | 56.6 | 32.8 | 87 | 19.2 | 71.6 | 55.6 |
| with GCN | 91.2 | 62.5 | 70.4 | 67.0 | 53.2 | 89.6 | 41.9 | 74.4 | 68.8 |

(a)                               (b)                               (c)

Figure 1:  (a) F1 score (in %) for *cross-database testing* on BP4D and UNBC using models trained on EmotioNet. (b) t-SNE visualisation of the features generated from two views to recognize AU25. (c) Proportion of samples with inconsistent prediction results from the two views.

**Reference:** [1] Li G. et al. Semantic Relationships Guided Representation Learning for Facial Action Unit Recognition. AAAI2019.

[Meta-Review · NeurIPS 2019]

This paper was reviewed by three experts in the field and received 667 recommendations. Based on the reviewers' feedback, the decision is to recommend the paper for acceptance to NeurIPS 2019. The reviewers did raise some valuable concerns among which are further interpretation of the dependency matrix, evidence of the mutual complement of the two networks, cross-dataset generalization, etc. These questions should be addressed in the final camera-ready version of the paper. The authors are encouraged to make the necessary changes to the best of their ability. We congratulate the authors on the acceptance of their paper!